# Smartphones as a Platform for Tourism Management Dynamics during Pandemics: A Case Study of the Shiraz Metropolis, Iran

Hadigheh Morabi Jouybari [1], Amir Ghorbani [2], Hossein Mousazadeh [3,*], Azadeh Golafshan [4], Farahnaz Akbarzadeh Almani [5], Dávid Lóránt Dénes [6,7,*] and Ritter Krisztián [6]

1 Department of Regional Economics and Rural Development, Hungarian University of Agriculture and Life Sciences, 2100 Gödöllő, Hungary
2 Department of Tourism, Faculty of Geographical Sciences and Planning, University of Isfahan, Isfahan 81746-73441, Iran
3 Department of Regional Science, Faculty of Science, Eötvös Loránd University, 1053 Budapest, Hungary
4 Department of Communication and Journalism, College of Arts and Social Sciences, Osmania University, Hyderabad 500007, India
5 Department of Tourism Management, Budapest Business School, University of Applied Sciences, 1149 Budapest, Hungary
6 Institute for Rural Development and Sustainable Economy, Department for Rural and Regional Development, MATE Szent István Campus, Hungarian University of Agriculture and Life Sciences, 2100 Gödöllő, Hungary
7 Faculty of Economics and Business, John von Neumann University, 6000 Kecskemét, Hungary
* Correspondence: hmosazadeh5575@yahoo.com (H.M.); dr.david.lorant@gmail.com (D.L.D.); Tel.: +36-204359609 (H.M.); +36-0209722833 (D.L.D.)

**Abstract:** During the past three years and with the spread of the pandemic, smartphones were the most important communication bridge between tourists and organizations; now more than ever, they are intertwined with the lives of tourists and destination management organizations. Although much research has been conducted in this field, the investigation of the effects of the pandemic on the technology and functionality of smartphones is one of the topics that has been less discussed. Therefore, the current research was conducted to determine the role of smartphones in tourism management dynamics during the pandemic. The research method was qualitative (content analysis, theme analysis), and 32 people participated in the interview process as a statistical sample. Then, the oral interviews were transcribed, and a thematic analysis was performed. For the analysis of the interviews, MAXQDA 2020 software was used. The results of the research indicate that smartphones were one of the most important platforms for tourism management dynamics during the pandemic, and in the event of a pandemic in the future, they can help contain the destruction to a great extent in their current position.

**Keywords:** destination management organizations; tourism technology; content analysis; thematic analysis



## 1. Introduction

The experience of the COVID-19 pandemic and its effects on the tourism industry have caused an indisputable shift in the structure of tourism management [1]. It could be argued that the most salient consequence of this pandemic over the past three years has been the lack of dynamism in the tourism management structure to face such unprecedented conditions [2]. However, the development of smart tourism may reduce the tourism sector's vulnerability to unexpected environmental threats [3]. During the pandemic, smartphones have become the focus of attention for tourists and destination organizations more than ever before [4]. While the use of smartphones in tourism management is not a novel concept, its changes brought by the pandemic era certainly warrant further exploration [5]. Due to the newly discovered and highly infectious COVID-19 variants, mainly unvaccinated groups and unequal tourism rises across locations can hamper vaccination prospects [6].

While tourism looks to be on the mend, the expansion of such new variants might stop it [7]. Currently, there are signs of a drop in the flow of tourism due to the reemergence of the COVID pandemic in Japan and the USA [8]. Now, smartphones are more integrated with the structure of tourism and the lives of tourists than before the pandemic, and it can almost be said that they have become a part of the lives of tourists and destination organizations [9]. As a consequence, communities and locations can be linked virtually with the help of new communication devices and applications [10]. As these phones are connected to social media throughout the world, they provide a chance for tourists and destination management organizations to disseminate their information to other users [11]. During the COVID-19 pandemic, with the tightening of restrictions, more tourists used smartphones and various applications and social channels. Smartphones even completely replaced physical travel, and the number of followers of travel bloggers increased greatly [12]. Tourists confined at home published the data of their past trips or used travel blogs as a tool for planning and choosing a destination. Therefore, smartphones have been utilized as a "psychological management tool". Additionally, as a planning tool for the new tourism platform or postmodern tourism era, it can be referred to as a "behavioral management tool". To do so, travel executives and specialists can apply new strategies to design new travel structures, such as single-tourist travel with long-distance privacy; hence, fewer travels with fewer travelers can protect nature and, as a result, develop sustainable tourism [13]. Therefore, the current research was conducted to focus on the functions of smartphones in tourism management during the pandemic. Although the use of smartphones in the tourism industry is not a new issue, the most important novel aspect of the current research is the investigation of the effect of the pandemic on the functioning of smartphones during the pandemic. Smartphones were not seriously used in tourism in countries such as Iran before the pandemic. Shiraz metropolis is one of the most important destinations for medical tourism, including cosmetic surgery and hair transplant tourism in the Middle East. Tourists who travel to Shiraz for laser cosmetic surgeries and hair transplants benefit from hotel and tour services. Many taxi drivers and tourism service personnel in Shiraz are fluent in English and communicate well with tourists at airports and passenger terminals [14]. In addition, due to the active cycle of tourism economics and the focus of medical tourism operators in the Shiraz metropolis on smartphones as a means of communication with tourists, this city was selected as the research area. On the other hand, Shiraz tourism officials intend to design tourism in the Shiraz metropolis based on smartphones soon.

Although the perpetual vital role of smartphones in tourism management is undeniable [12]; an overwhelming number of studies have been conducted to examine the impact of the pandemic on tourism [15], but few studies have studied the pandemic's impact on tourism technological behaviors. In an attempt to bridge this gap, this study aims to investigate the alternations that COVID-19 has brought about in tourists' purposes and tourism organizations concerning the use of their smartphones.

## 2. Literature Review

### 2.1. Tourism Management Dynamics and Smartphones during COVID-19

One of the most important challenges of tourism management during the pandemic in the past three years was that destination planners and tourists did not foresee an epidemic, and what strategy should be adopted in the case of such a situation was not foreseen [16]. One of the main tasks of destination managers and planners is to identify future trends and threats. Additionally, identifying potential opportunities caused by environmental factors can make the structure of tourism management more flexible in adapting to turbulent times. The dynamic structure of tourism management in this situation can reduce the severity of the damage by "controlling the destruction". Additionally, focusing on modern technology and smart structure leads to the emergence of new businesses that adapt to turbulent conditions. On the other hand, smart platforms are in the spotlight in this situation with user change [17]. The ability to use technology to benefit from existing conditions is

one of the effects of tourism management dynamics [18]. With the spread of the pandemic, smartphones have become the focus of attention of consumers and service providers to reduce unnecessary calls and cut off human communication [19]. The relationship between the audience and smartphones increased to such an extent that the concept of smartphone addiction was raised [20]. Verma et al. (2022) argue that the interaction between tourists and smartphones has taken a new form [21]. In recent years, much research has been published regarding the role of smartphones in tourism. However, the functionality of smartphones during the pandemic is still a vague issue. Perhaps the functions and roles of smartphones in turbulent times such as pandemics are vague points in the literature of this field, and it is impossible to comment on them decisively.

On the other hand, smartphones greatly changed the classic and modern organizational structure during the pandemic era and created an organization based on intelligence and technology according to the current conditions. This new organization was not limited to a physical organizational space, was based solely on smartphones, and continued its communication with tourists. This form of organization had the necessary dynamics to operate during the pandemic and was built according to the ideas of the postmodern organizational school [22]. Therefore, smartphones have assumed a more vital role than in the past [23]. Additionally, the roles of human resources and their functions are widely affected by these new processes [24]. Many employees lost their jobs, and new jobs were created [25]. Now, three years after the outbreak of the pandemic, organizations are often two-dimensional. The first dimension is related to the physical organization, and the second dimension is related to the virtual personality of the organization, which is managed by a person with the title of virtual affairs manager. At this level, smartphones promoted POPAI in this field by establishing the virtual dimension of the organization and maintaining the relationship between the organization and the tourist. The most important function of smartphones during this period was that after doubt in the initial months, the communication between destination organizations and tourists remained connected.

### 2.2. Smartphone Trends in Tourism Management and Tourist Behavior

Generally, internet usage in the tourism context is divided into two stages: first, from 1991 to 2002, when the internet was pursued by the whole tourism industry; second, from 2002 to the present, when tourism markets have concentrated on conviction and authorizing the consumer by using (more recently ubiquitous) mobile devices, specifically smartphones [26]. At this stage, where smartphone adoption has amplified the significance of consumers more than ever and has reshaped tourism and its business market, realizing consumer behavior toward using mobile technologies due to their competency in access to the internet, GPS, communication channels, and photography applications seems essential to succeeding in the current business industry [27]. Mobile technologies in tourism have been designed and produced to provide electronic information as guidance and help tourists in decision-making [28]. Indeed, traditional mobiles were only capable of calling and texting [29]. However, presently, the most up-to-date mobile apps available through smartphones have a great impact on both the demand side (customer) and the supply side (market) in tourism [26] since they equip travelers with wireless, instant, and prevalent internet access, thus enabling them to send and receive any helpful information globally. The growing impact of sustainable marketing operations in the tourist sector results from technological advances and the acceptance of smartphones [30]. Dickinson et al. [31] define the smartphone as a potent device for tourists because of its ubiquity in the interchange of social and on-site information, as well as connecting them to remote information databases. For instance, smartphones are currently applied for pretravel planning, decision-making, shaping expectations, and pretravel previsions; they are further used for navigation, connectivity, instant/short-time decisions, and location-based data exchanges during travel [32]. The estimated number of smartphone subscribers in 2021 was over 6 billion worldwide, and it will likely rise by several hundred million in forthcoming years [33]. The perpetual and endless variety of tourism apps and their ever-growing users transformed tourism

activity from a location-based network to a distinct network [31]. In the same way, in the tourism industry, smartphones and travel-related applications have become very powerful and noticeable for searching for the needed information, accordingly leading to tourists' decision-making [27]. Based on Petrescu and Bran [26], the online shopping frequency of tourism-related services ranks third among other groups. Tourists who look for travel information during their pretravel time usually continue searching during the trip [34]; thus, tourists' information search behavior is considered a continuous process [35]. However, there is still insufficient knowledge for understanding serious matters such as information classifications searched by tourists through smartphones and the differences between information searched before and during the traveling period [11]. It is confirmed that tourists may alter their behavior during the trip and modify their initial plans due to the information they acquire through their smartphones. Similarly, ref. [36] it is empirically indicated that tourists' behavior is influenced by using information technology during their journey. The increasing use of smartphones, together with other recent developments, has resulted in the birth of smart tourism, which aims to increase the acceptance of new ICT forms [26]. Smartphones have become popular in the tourist and hospitality business as a convenient method for travelers to find information and book accommodations [37]. To date, tourists' concern has been primarily on particular matters such as booking plane tickets, booking hotels, and time management during trips [38]. Likewise, studies on the travel usage of smartphones have mostly focused on specific subjects, such as the advancement of mobile applications [39], the admission of smartphones as a public ICT tool, and the effect of using smartphones on various facets of tourists' experience [40]. The outbreak of COVID-19 has been identified as one of the most destructive occurrences in global history, affecting the tourism industry due to the prevention of human contact and movement [41]. Therefore, as a result of ongoing travel limitations and reduced travelers' conviction, the number of tourists declined by 87% in January 2021 compared with 2020, showing an unusual plunge of 73% throughout tourism history [42]. However, the restrictions caused by this calamity can be observed as an opportunity for travel destinations to advance by presenting their attractions and offering their product and services through a diverse number of hi-tech travel tools [43]. Sañudo et al. [44] showed that following the recent coronavirus pandemic and its criteria related to global lockdown, the usage of smartphones in many countries, including Denmark, Italy, Spain, the Netherlands, and the UK, grew dramatically. In this respect, the use of travel-related applications existing on smartphones changed to a great extent. Thus, the hospitality and tourism industry needs to be prepared to proactively overcome unexpected circumstances by gaining comprehensive knowledge of how to meet travelers' demands in this critical period.

*2.3. Tourism Postmodern Organizations and Turbulent Times*

(To read more about the organization from a postmodern point of view, he authors recommend "Organization theory: Modern, symbolic, and postmodern perspectives", Oxford University.)

Organizational theorists examine the organization from three points of view: modernist, interpretive symbolic, and postmodern [22]. The "modernist view" considers the organization as an independent objective entity and takes a positivist approach to produce knowledge. The "interpretive symbolic view" sees the organization as a society that remains stable through human relations and scrutinizes how to create meaning to make facts understandable for those who participate in maintaining them. "Postmodernism" creates a kind of "healthy skepticism" of any "dominant theory" and permits one to try anything completely differently. It seems that the organization in its postmodern formulation is more compatible with the nature of tourism because it provides the necessary dynamism and flexibility for suppliers and applicants of tourism services [3]. Additionally, the organization in this formulation changes according to the technology of the current time and the conditions of the surrounding environment of the organization. In this view, the organization is not just a physical space in a certain place; rather, it can change its

nature according to the circumstances. This approach is compatible with the transition from physical to virtual organization and is based on smartphones in turbulent times such as pandemics [11].

## 3. Materials and Methods

### 3.1. Study Site

Shiraz metropolis is located in southeastern Iran and is the capital of Fars province (see Figure 1). Its altitude ranges from 1480 to 1670 m above sea level in various districts of the city. Shiraz, known as the cultural capital city of Iran, has a history of more than 4000 years, is known as a city of culture, literature, and art, and has great potential for cultural and art tourism. Shiraz is a sensible destination for medical tourists in the Persian Gulf countries and internationally and has been active in attracting medical tourists from neighboring countries. According to Shiraz University of Medical Sciences, the low cost of health care and the acquisition of global rankings (for example, third place in organ transplantation globally) has made this city the center of medical tourism, including cosmetic surgery tourism, hair transplant tourism, and dental tourism [45]. In future planning, one of its goals is to become a medical tourism hub in the Middle East and Asia. Shiraz is renowned as a city with several world-class cultural heritages from a cultural and tourism standpoint and is the host of the tomb of Persian-speaking poets (such as Hafez and Saadi). Above all, Shiraz has various world heritage sites, such as the Takht-e-Jamshid and Persepolis complex palaces, attracting over a million tourists all over the world yearly [46]. Over the past years, the metropolis of Shiraz has developed its tourism based on smart tourism. For this reason, during the outbreak of the COVID-19 pandemic, smartphones have been used to reduce the severity of damage to tourist destinations.

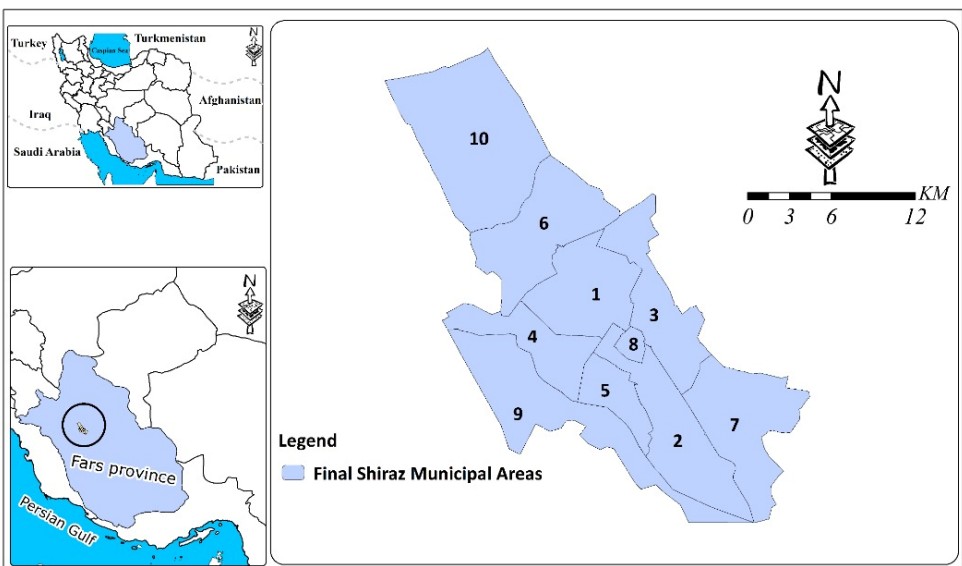

**Figure 1.** Location of Shiraz metropolis.

### 3.2. Data Collection and Analysis

The statistical sample of the current research consists of two groups, A and B, each group consisting of 16 members as participants. Group A includes smartphone users of Shiraz tourism organizations who managed the virtual organization during the pandemic. Group B also consists of 16 Shiraz travel bloggers, whose content had the most views during the pandemic. In qualitative methods such as theme and content analysis, the sample size is usually small. In this research, due to the limited size of the sample, the participants were selected using the census method, and 32 people were interviewed. In the second stage, the process of interviews began, and the time of each face-to-face interview was approximately 30 min. The interview questions consisted of 5 questions

that are presented in the appendix of the research. The research data were collected cross-sectionally in 2021 and the early months of 2022. All conversations were first recorded and then transcribed. After completing the interviews, all of them were transcribed. In content analysis, transcribing the text of the interviews is one of the most important steps to obtain research codes. After extracting the main themes, subthemes were assigned to related subgroups. In the third step, themes and subthemes were extracted from the transcription text and analyzed using MAXQDA 2020 software. (Table 1).

**Table 1.** Research participants.

| R | Group A | Number | Participant as Sample |
|---|---------|--------|-----------------------|
| 1 | Smartphone users in destination management organizations | P1 P2 P3 | User in a five-star hotel |
|  |  | P4 P5 P6 P7 P8 P9 P10 P11 P12 P13 P14 | User in a travel agency |
|  |  | P15 P16 | User in a four-star hotel |
| **R** | **Group B** |  | **Tourists** |
| 2 | Bloggers | 16 | Travel bloggers during the pandemic |
|  | **Total** |  | 32 |

## 4. Data Analysis

After transcribing the interviews, the main themes and subthemes were extracted, the results of which are presented in Table 2.

**Table 2.** Results of content analysis.

| Themes | Subthemes | Frequency |
|--------|-----------|-----------|
| Tourism management dynamics during pandemics | Identify trends early, Design proactive strategies | 24 |
|  | Identify potential opportunities from changes and threats | 23 |
|  | Continuation of tourism businesses during the pandemic | 22 |
|  | Adjusting human resources and the emergence of new tourism businesses | 20 |
| Psychological impacts | Expanding the network of tourist friends | 21 |
|  | Increasing the resilience of tourists locked up at home | 22 |
|  | Changing preferences of tourists during the pandemic | 24 |

**Table 2.** *Cont.*

| Themes | Subthemes | Frequency |
|---|---|---|
| Smartphone marketing | Perceived ease of use of smartphones in marketing tourist sites | 21 |
| | The tourist is more exposed to marketing activities | 20 |
| | Smartphones are available to tourists anytime and anywhere to promote marketing activities | 19 |
| | In smartphone marketing, the tourist can quickly connect with the source of the message (hotel, travel agency, fast food services, etc.) | 22 |
| Travel planning | Familiarity with different potentials of tourism in the destination and new entertainment | 23 |
| | Better choice of travel destination by watching high-quality videos and images, and acquiring more information through social media | 21 |
| | Communication with tourism service providers at the destinations | 22 |
| | Make payments, reservations, and other things before traveling | 19 |
| Postmodern organization | | 24 |
| | Breaking the concept of organization as a physical space | 23 |
| | Virtual organization is the second dimension of physical organization | 22 |
| | Adjustment of human resources | 24 |
| | Creating new jobs based on technology and current conditions | 22 |
| | Increasing communication and two-way interaction between the tourist and the destination organization | 18 |
| | Virtual organizations are open in all conditions | 23 |

*Final Model Design Based on the Findings*

From an organizational perspective, the organizational boundaries between the tourist and the top level of the organization (manager of a large hotel or manager of a travel agency) collapsed, which is one of the main signs of postmodern organizations [47]. Therefore, according to the MAXQDA analysis, the final model of research is presented.

## 5. Results

The research results indicate that smartphones played a vital role during the pandemic and caused positive changes in the structure of tourism management. Tourism has now, to some extent, achieved the necessary preparation for conditions such as a pandemic or other turbulent times; if the conditions repeat, the amount of damage will be less. After the initial shock due to the outbreak of the pandemic in 2019, tourists and destination organizations were able to communicate with each other again using smartphones. Organizations took a new form as a "virtual dimension", and there were extensive changes in tourists' preferences. In the final part of the research, the dimensions of the model will be examined separately.

*5.1. Smartphones as Part of Postmodern Tourism Organization and Tourism Management Dynamics*

Virtual tourism organizations designed according to current issues of tourism have different natures in terms of organizational and physical structures [3]. In the early months of the pandemic, many of the destination management organizations (DMOs) that were physically shut down were transferred to virtual organizations in smartphones, and smartphones increased in popularity among tourists and organizations [48], as tourists had continuous two-way communication with them. At the beginning of 2023, smartphones will become the most important part of the life of DMOs and tourists and have already broken the classic organizational structure. According to the research model, the most important advantage of this type of organization in the tourism management cycle is the survival of the organization in conditions of pandemics. Adopting adaptive and preventive strategies according to environmental conditions, equipping destination managers and planners with new management techniques according to technology to benefit from existing conditions, identifying potential opportunities due to changes, and finally using new technologies to obtain the maximum in business are the most important functions of smartphones in tourism management dynamics [18]. Additionally, postmodern organizations are more in tune with the latest technologies due to their open relations with their environment. From the perspective of organizational conflict, the attitude toward conflict in postmodern organizations is a constructive phenomenon. For exactly what smartphones bring to tourism organizations and tourists, see Figure 2. Smartphones have shattered the structure of the classic relationship between tourists and DMOs and have turned tourists and the organization into friends who are constantly in touch. From the perspective of organizational postmodernism, pandemics can be an opportunity for tourism organizations to make themselves a more flexible concept called "breaking the foundation" in the theory of organizational postmodernism. The organization is no longer an inflexible machine, but a painting screen run by an art director [49]. Comprehensive access to and large investment in the development of VR indicate its potential relationships in the development of tourist destinations [50]. Concepts such as digital tourism, virtual tourism, and smart tourism are the results of the information and technology era [51]. The technology acceptance model (TAM) and stimulus-organism-response (SOR) framework are mostly used in VR and AR tourism studies [52]. VR spurs tourists to daydream about tourism suggestions before experiencing them on destination premises [53]. Tourism can strengthen the sense of empathy among tourists in times of crisis and disasters [54]. Observing tourism videos and their impact on tourist empathy [55], the question "how to activate empathy and online platform operators" [56] has been studied. Three factors were raised by the participants as compatibility with the pandemic theme, the most important of which was "A virtual trip to Shiraz without physical contact with the host community," a 35-year-old woman (a smartphone user in a four-star hotel) said:

> "COVID-19 has prevented tourists from traveling to Shiraz. In turn, it has disrupted tourism-related businesses. Starting a virtual trip and sharing clips from Shiraz can relaunch the relevant businesses and ensure the health of tourists."

Based on our results, it was suggested by all of the respondents that virtual travel during the COVID-19 pandemic could be a practical proposition. It also seems that sharing Shiraz's videos and images on social media makes it possible to strengthen empathy and unity among tourists during the quarantine period. Similarly, a 29-year-old man (a smartphone user in a travel agency) said:

> "As the mayor of Shiraz explicitly stated during the COVID-19 pandemic, "empathy and unity should be the priority of Shiraz tourism authorities," and, on the other hand, during quarantine, the feelings of empathy, cooperation, and mental health of tourists decreased. In my opinion, by creating Shiraz websites and virtual tours, we can improve the damage."

A 28-year-old man (international tourist) stated:

"I have traveled a lot to Iran; most of my trips have been to Shiraz. My trip to Shiraz was due to my best friends, whom I found through social media, especially Instagram. Before starting COVID-19, I traveled to Iran every summer. Now my friends who live in Shiraz always send me photos and videos of Hafez and Sa'adi. The last time I saw Shiraz on a virtual trip was through a video call. My smartphone gave me by my friends. Our relationship has become much more intimate due to quarantine conditions and travel rules. My smartphone played a key role in the formation of these friendships, and in this situation, I needed to communicate with them mentally. Thank you, my smart friend."

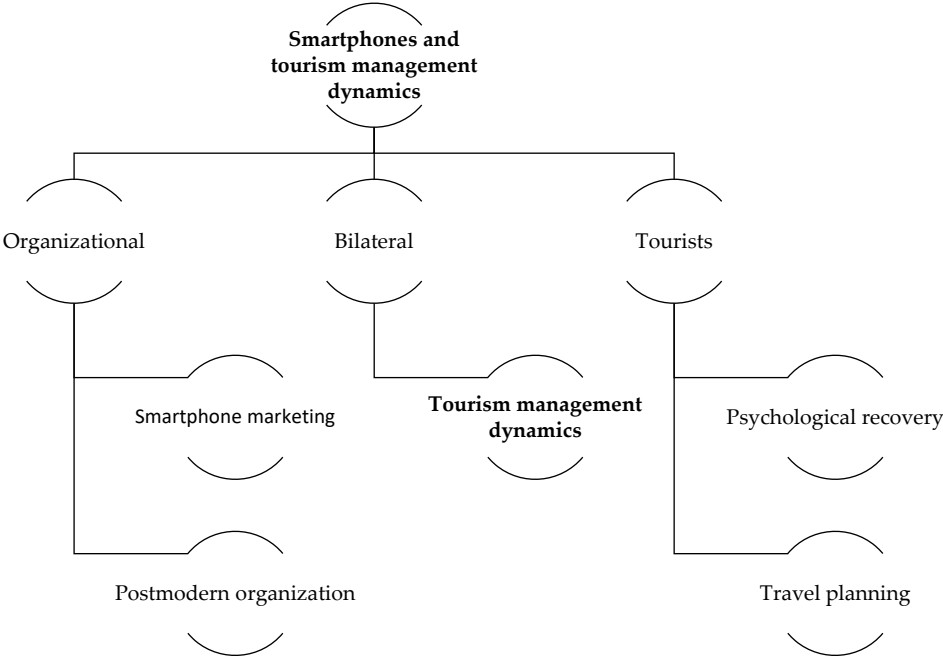

**Figure 2.** Final model design.

## 5.2. Psychological Recovery

It is fully acknowledged that smartphones have become an inextricable part of tourism activities due to their vital role in obtaining and disseminating information [57]. As an innovative and growing ICT tool, the smartphone has advanced the quality of travel basis and turned it into a ubiquitous, dynamic, and fluid context [58]. According to Rodríguez-Torrico et al. (2019), mobile phones have become superior technological tools for tourists in recent decades [59]. This has a great impact on travel patterns as well as the behavior of travelers [60]. After the prevalence of COVID-19, tourists ceased traveling for a long period due to the global lockdown and the perceived risk of becoming infected [61]. This leads to new tourists' behavior, such as reducing activities with face-to-face contact and considerable care about sanitation and cleanliness [62]. In addition, on March 18th, 2020, an issue was released by the Department of Mental Health, WHO, regarding the psychological precautions for maintaining mental health during the pandemic [63]. In a tourism context, the association between tourism activities and mental health improvement has long been verified [9,10]; however, due to the disruption of tourism mobility during the COVID-19 outbreak, tourists were deprived of this privilege. Nonetheless, according to the results of the current study, people could take advantage of their smartphones to diminish the psychological effects of the crisis to some extent. This could assist tourists through the expansion of the virtual networks between their traveler friends and sharing their past travel diaries. Additionally, the resilience of locked-up tourists has greatly increased as a result of virtual communication relating to travel matters through mobile networks and applications. Moreover, this led to altering the preference of tourists during the pandemic era, which contributes to their health security at this time. Similarly, Lu et al. (2022) support

these research outcomes, asserting that utilizing virtual tourism through smartphones as an amusement tool in the time of the most recent pandemic can endow people with an immersed tourism experience inside their place and assist their tolerance during the lockdown. It is believed that such novel practices are being continued by tourists in post-pandemic times for various purposes [64]. As the world wrestles with the facts of COVID-19, there is an occasion to rethink precisely what tourism will look like in future decades [59], and there is ground to revise world economic value chains and the particular role of tourism as a vector in the outbreak of pandemics [65]. Post-COVID-19 situations may help reimagine "future-forward" worlds. Exactly what is helpful is a "future-back" strategy [65]. As we know, the absolute impact of tourism and tourists make the future of tourism [66]. The value and significance of virtual communities of practice in palliative care are becoming increasingly clear [67]. The COVID-19 period can have immediate or long-term mental health consequences [64]. The main mental health concern has inflated the amount of worry or anxiety reflected in public mental health terms as the COVID-19 epidemic has spread rapidly all over the world [68]. The situation can accelerate new mental ailments and intensify the earlier present disorders [69]. COVID-19 outbreaks have also been shown to damage the morale and mental health of those who already have mental problems [70]. Quarantined people feel apathy, violence, sadness, fear, rejection, despair, insomnia, harmful substance use, and loneliness [40,57]. (Figure 3). A 31-year-old man (an Iranian tourist who lives outside of Iran) stated:

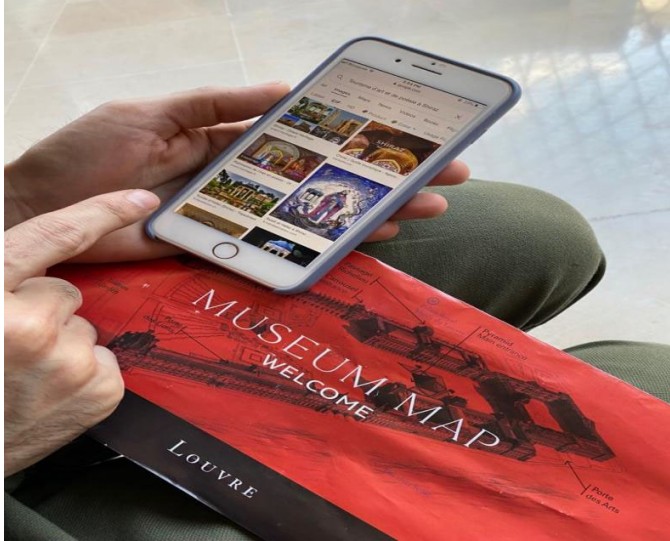

**Figure 3.** A foreign tourist browsing the memories by watching online the tourist attractions of Shiraz.

> "I am an Iranian, and I live with my family in Europe. I recently visited the Louvre Museum in Paris. I started my visit due to my dependence on Iran and the Iranian collections that are in Level 0 and section 308 of the museum. During the visit, I felt very homesick, and my trip to Fars was associated with me. I quickly searched for the cultural and historical places and the tomb of the Persian-speaking poets of Shiraz on my smartphone. It was like a virtual journey for me that made me feel good and lessened my nostalgia."

*5.3. Smartphone Marketing*

Smartphones have been used as an integral means of marketing in recent years, with the power of changing the mind of prospective consumers from negative to positive toward purchasing goods or services [71]. Smartphone marketing can be regarded as a two-way interaction between DMOs and their customers [72]. Using smartphones, tourists have ubiquitous access to all kinds of travel-related information [73]. The lockdown period revealed the value and necessity of smart technologies in helping people cope with the

social and psychological chaos of the crisis [74]. While the main analysis stream on tourism technologies has examined the adoption of smartphones and web-based technologies, the potential of destination marketing through virtual technologies is still to be fully investigated [75]. Similarly, recent studies on COVID-19's effects on the tourism business emphasize the need to change the ways of practicing the tourism business and recommend that tourism locations will have to update their current business models in the future [70,71]. In the same way, the current results show four factors of smartphone marketing that helped tourists deal with isolation. First, smartphones offer tourists perceived ease of use to access travel-related websites, which is a significant marketing tool. Likewise, travel websites such as TripAdvisor offer a great amount of reliable information because customers believe their counterparts more than brands [75]. In addition, tourists are more exposed to marketing activities via smartphones, as they are accessible to tourists anytime and anywhere they need. This phenomenon acts as a mutual benefit for tourists and tourism organizations as a promotional marketing factor. Moreover, by smartphone marketing; tourists can instantly connect to the source of information such as travel agencies, hotels, food services, etc., and check their authenticity. As a consequence, in the post-pandemic era, tourists search for more information than ever before, and new development strategies based on cutting-edge technologies seem necessary specifically for tourism market research. Social media is an essential tool for tourism destination development. The sense distributed in these programs has a fundamental role in tourists' decision-making and tourism management [76]. A film can present a large display promotion of a tourist destination [77]. In response to the growth of social media programs, advanced tourist choice, and technological development, the value of consumer-generated content (CGC) stretches to develop organizations marketing their destinations, products, and services to tourists [78]. A 35-year-old man (a smartphone user in a travel agency) stated the following:

> "Social media platforms in times of crisis are the best way to offer tourist attractions. In this case, a robust database is provided for tourists who can check it at any time using their phones. In this way, tourists can also publish their experiences and strengthen their empathy and mental health by reviewing memories."

Furthermore, a 48-year-old woman (a smartphone user in a five-star hotel) said:

> "By creating various tourism applications in Shiraz, the tourism capabilities of the city can be provided in detail in different languages. In recent years, the Tourism Organization has launched the virtual tourism portal of Shiraz with the use of panoramic images and virtual tours, videos, photos, maps, and descriptions (visit www.fafarschto.ir (accessed on 25 June 2022))."

Additionally, a 29-year-old woman (a student and tourist) stated:

> "Recently, joint meetings of Iranian and European tour operators, especially Hungarian ones, have been held with the help of Shiraz Municipality. There are very close ethnic, cultural, and racial commonalities between the Iranian people and the tribes living in the Hungarian city of Jászberény. Many of them want to travel to Shiraz. As a tourist in Hungary, I have always tried to show them the potential of tourism through the websites available for virtual tourism in Fars and Shiraz provinces. In my opinion, smartphones can be very effective as a tool for marketing tourism compatible during a pandemic in Shiraz." (Figure 4).

According to the findings of this section, the use of smartphones in the Shiraz metropolis can strengthen the marketing of the host community [79]. Consistent with these results, Prideaux et al. [75] also believe that insights learned during COVID-19 can help prepare global tourism for the economic revolution that is needed.

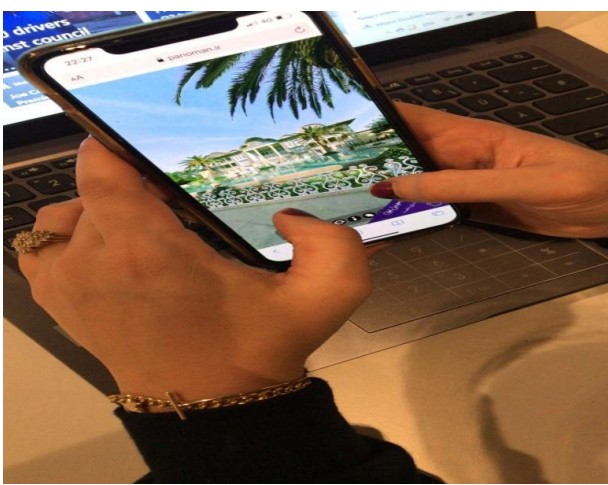

**Figure 4.** Smartphone marketing and virtual tourism of Shiraz.

*5.4. Travel Planning*

Currently, tourism directors use smartphones to interact and familiarize their customers with tourism destination offerings. Tour executives use social media to obtain notified decision-making about tourists' destinations [80]. Social media is recognized as an essential information reference that affects tourists' travel choices [81]. Social media also plays a significant role in tourists' purchase decisions [82]. Modern travelers' intention to begin a trip is determined frequently by suggestions from friends and family, online references and observations, and, to some extent, knowledge given by a third party [83]. One of the most important elements that visitors consider while planning a trip to a certain location is the dependability of data references. Trust is vital in online tourist marketing since it encourages them to buy. Almost all travelers utilize ICT and cell phones to obtain information about tourist sites, hotels, and accessibility [84–87]. Smartphones, which have recently gained a foothold in tourism, have the potential to introduce the potential of tourism to tourists. For example, using the Internet and social media can show its capabilities to others. Due to their portability and access to the Internet, smartphones have emerged effectively to meet the needs of tourists. Smartphone systems are important tools that move tourists to the virtual world; in this regard, the role of information and communication technology, smartphones, and their tools for travel planning is very significant [80,88–90]. Based on the results of this study, using smartphones by tourists during the quarantine period has helped tourists to become more familiar with the diverse potential of tourist destinations and the new entertainment they offer. Tourists can also have a better choice of travel destination by watching high-quality videos and pictures and obtaining a great amount of information on social media channels.

In this regard, a 34-year-old man (a tourist) said:

"After essential travel items, such as passports, the smartphone ranks first in "what to take with you on the trip." The last thing I had before going to bed was always a mobile phone so that I could make the next day's travel plans. During the trip, the most common use of smartphones was to take photos, post them on Instagram and WhatsApp, and then use the map. Many large digital companies, such as Google, Facebook, and National Geographic, have a section called travel. In my opinion, applications such as MyShiraz or Shiraz travel can be suggested for planning a trip to Shiraz."

**6. Conclusions**

The metropolis of Shiraz is known as the largest city in southern Iran, and tourism is one of the economic and entrepreneurial pillars of this city. Shiraz has good infrastructure in all dimensions of the field of tourism and is a well-equipped destination for international

tourists in various fields. On the other hand, a large number of citizens are employed in the private sector of tourism in this city. Shiraz was one of the tourist destinations that suffered the least damage in the tourism sector during the COVID-19 pandemic. Many of the important medical tourism centers of Shiraz metropolitan city, after passing through the initial shock period caused by the pandemic, provided the necessary training to the patients through smartphones to restart their actions. During 2021, many tourists traveled to Shiraz for hair transplants, cosmetic dentistry, slimming and prosthetic surgeries, and other medical procedures, while they had already received instructions on how to travel safely in pandemic conditions via smartphones. Smartphones contributed greatly to the development of medical tourism in Shiraz during the pandemic. Smartphone-based "virtual organizations", as well as travel bloggers, played an important role in this regard. After passing through the initial shock and curbing the destruction of tourism, focusing on technology was identified as the priority. Two-way communication between managers and tourists was formed, and tourists were continuously instructed to safely travel during the pandemic through popular smartphone programs. Gaining opportunities from environmental threats and adapting to turbulent times is one of the characteristics of tourism management, dynamics, and destination management organizations in the Shiraz metropolis that were able to implement this concept by using smartphones. Shiraz is now a successful example of the dynamics of tourism management in the pandemic era and can improve its share of tourism income shortly. Therefore, from the point of view of organization theory, it can be concluded that destination organizations in the modern sense are no longer responsive to the needs of the tourism industry in turbulent times such as pandemics. On the other hand, bloggers and travel influencers produced suitable content for tourists beyond smartphones during the recession era. For many tourists with pandemic control, their first destination was Shiraz because many virtual friends were waiting for them at the destination. Creating a network of virtual tourist friends and shaping tourist preferences were the most essential functions of travel bloggers in the Shiraz metropolis, so it can be said that smartphones made the current tourists of Shiraz travel more informed and sustainable compared to tourists before the pandemic. A case study of Shiraz metropolis, although on a small scale, suggests that tourism is now more prepared to deal with turbulent times compared to the pre-pandemic period. The results of the present study have been carried out on a case-by-case basis in the metropolis of Shiraz, and the generalization of its effects, in general, requires large-scale research. Therefore, it is suggested that researchers study the functions and roles of smartphones in the tourism management literature from a newer perspective, such as the effect of smartphones on the adjustment of human resources of destination organizations, the role of smartphones in managing the emotions of tourists in turbulent times, and the role of smartphones in the development of sustainable tourism, which are considered suitable subjects in this field. Additionally, developing a crisis control model for tourism businesses in turbulent times such as epidemics is a suitable idea for future research that requires more extensive studies.

**Author Contributions:** Conceptualization, A.G. (Amir Ghorbani) and H.M.; Methodology, A.G. (Azadeh Golafshan) and F.A.A.; Software, A.G. (Azadeh Golafshan) and H.M.J.; Validation, R.K., A.G. (Amir Ghorbani), F.A.A., D.L.D. and H.M.; Formal analysis, A.G. (Azadeh Golafshan), H.M., R.K., A.G. (Amir Ghorbani), F.A.A. and H.M.J.; Investigation, H.M., D.L.D. and R.K.; Resources, H.M.J., H.M. and F.A.A.; Data curation, R.K., D.L.D. and A.G. (Amir Ghorbani); Writing—original draft preparation, H.M., A.G. (Azadeh Golafshan) and A.G. (Amir Ghorbani); Writing—review and editing, R.K., F.A.A., D.L.D., H.M. and H.M.J.; Visualization, H.M., A.G. (Amir Ghorbani) and A.G. (Azadeh Golafshan); Supervision, H.M., R.K. and D.L.D.; Project administration, R.K., D.L.D. and A.G. (Amir Ghorbani); Funding acquisition, R.K. and H.M.J. All authors have read and agreed to the published version of the manuscript.

**Funding:** This research was created and supported by the Hungarian University of Agricultural and Life Sciences (MATE), Doctoral School of Economic and Regional Sciences and Stipendium Hungaricum Scholarship.

**Informed Consent Statement:** Informed consent was obtained from all subjects involved in the study.

**Data Availability Statement:** Not applicable.

**Conflicts of Interest:** The authors declare no conflict of interest.

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
