# Peer review of "Smartphones as a Platform for Tourism Management Dynamics during Pandemics: A Case Study of the Shiraz Metropolis, Iran"

_sustainability, doi:10.3390/su15054051_

Round 1
Reviewer 1 Report
REVIEWER'S DETAILED SUGGESTIONS
ABSTRACT:
The abstract MUST be structured as follows:
1. Purpose (mandatory)
2. Design/methodology/approach (mandatory)
3. Findings and Originality/value (mandatory)
4. Research limitations/implications (if applicable)
5. Practical implications (if applicable)
6. Academic implications/Further research (mandatory)
INTRODUCTION:
Introduction MUST discuss the following areas:
1. Background (ontology / the existing facts, as so liable to be research academically)
2. Research Objective (MANDATORY); and MUST be synchronised with the research method, theoretical framework, discussion and conclusion.
3. The reasons why the research questions liable to be examined
4. Novelty or existing gaps
5. Expected benefits and contribution
LITERATURE REVIEW:
This section MUST discuss the following areas:
1. Previous studies (mandatory)
2. Concept and / or major theory used in the articles
RESEARCH METHOD:
This section MUST discuss the following areas (but not limited):
1. Selected location (Why they are selected case significant?).
2. What is the sampling procedures?
3. Any ethics clearance has been obtained for this study?
4. How did you discuss the analyzed qualitative data in the discussion/finding section?
5. Problem statement / purpose of study
FINDING & DISCUSSION
This section MUST discuss the following areas:
1. Finding based on the data gathered and answering all research questions
2. Discussion: reflections of the "current study" compared with the "previous studies"
CONCLUSION
This section MUST discuss the following areas:
1. Researcher's view why the case is interesting to be investigated
2. Write conclusions based on the research questions
3. Research Implications and limitations
4. Suggestion for further research related to this finding
ENGLISH ISSUE
- The English needs to be proof-read
Author Response
Hi dear reviewer:
Thank you for taking the time to review the manuscript for our research team. We believe that your valuable comments can help improve our manuscript. So we carefully edited them according to your comments. In total, the manuscript was rewritten according to the comments of 4 respected reviewers. We hope our edited manuscript has the quality you need.
sincerely yours

Reviewer 2 Report
The article deals with an interesting topic, but there are a few things to modify and improve.
More important observations and remarks:
I have the impression that the article was written quite a long time ago and needs updating - it can be seen, for example, by the current variant of the virus (according to you - Omicron) and by not the latest literature. I suggest to update it.
Due to the very small sample size, this can only be considered as a contribution and inspiration for further research. Due to the size of the sample, these studies only indicated areas/topics for further in-depth research. Drawing far-reaching conclusions from the analysis of the answers of 32 very different people is rather frivolous.
What's more. The topics of these studies can (should) be limited rather to medical tourism, because the authors repeatedly emphasize the development of mainly this type of tourist services in this city. BUT - medical tourists have completely different expectations, preferences and behaviors than, for example, those who came to visit monuments or sunbathe on the beach. These issues (and a few others) should be included as limitations, which in principle have not been indicated in the article at all.
What is the purpose of dividing the city into 10 zones (Figure 1)??? I feel it is unnecessary.
Another thing is that the research results can only be limited to the city of Shiraz. This, in turn (assuming that mainly medical tourism is developing in this city, which cannot be implemented via a smartphone) indicates that the area of research has not been well-chosen.
The authors assure that these studies are innovative and fill a certain research gap. But reading Part 5 of Results and Discussion I have a completely different feeling. In this part there are a lot of theories and references to research by other authors (I do not consider this a mistake, because it is part of the discussion). But unfortunately, it looks as if the results of your research are only a confirmation of what is already investigated by other researchers and described (which you just point out in part 5).
Many articles on a similar topic have been published in recent months. Here the latest items are from 2021 (and that's just a few). Maybe it's worth including the newer ones in the discussion?
Other remarks were included in the text in the form of comments.

Author Response

(The authors gave the same response as above.)

Reviewer 3 Report
I am not sure where to begin. I am not sure this paper brings anything new to the academic literature. As I was reading the paper, it felt as if the authors were suggesting that the use of smartphones was a new concept in tourism, when in reality they have already long been a topic in the literature. The authors spend very little time in the results section dealing with the data, and in fact, move their data to the discussion section. Why define tourism on page 3? Who is the audience? The literature review was poorly written. Please see my handwritten comments for more details.

Author Response

(The authors gave the same response as above.)

Reviewer 4 Report
Dear Authors,
Thank you for the opportunity to read your work. My suggestions are the following:
1-Consider using keywords that are concise and germane to the topic. I doubt that MAXQDA in itself is deserving of a spot in the keywords section. Additionally, "tourism management compatible with pandemic conditions" can be broken down into several useful individual keywords.
2-Line 105: what other concerns do you refer to?
3-Lines 111 and 112: would you happen to have figures for Iran?
4-Line 227: I imagine that what you had in mind was "health protocols" instead of "health tips."
The article can be improved after revisions in presentation are applied. While I understand the logic behind having two groups (tourism officers and tourists), I do not think that this logic was clearly presented. Would it help if the groups' answers are placed side-by-side?
You can definitely build on the post-modern era argument by identifying the challenges and threats to tourism. I find it difficult to imagine, both as a scholar and tourist, how tourist enterprises can monetize the digital. Perhaps you can shed some light on this.
I recommend that the revised version be extensively edited. I understand that you and I, we, are not native speakers of English; this suggestion does not take away from your effort and scholarship.
Again, thank you for the privilege of reading your work. I look forward to reading a revised version.
Author Response

(The authors gave the same response as above.)

Reviewer 5 Report
The subject studied in the article is quite interesting, but once analyzed I want to give the authors some recommendations that they should take into account:1. The methodology used to analyze the data collected has not been justified.
2. Dates of data collection are not shown.
3. The importance of medical tourism in Shiraz has been commented several times and you do not take it into account when carrying out the empirical study.
4. The results obtained do not offer much information.
5. The conclusions are very brief.
6. No limitation found when making the article is mentioned.
7. There are no future lines of research.
Author Response

(The authors gave the same response as above.)

Round 2
Reviewer 3 Report
Still some English grammatical issues. See sentence 3 and 4 of the first paragraph of this paper, for example. Very long paragraphs, which makes the paper harder to read; harder to keep the reader's attention. "Although the functions of smartphones are not a new topic in the literature on tourism management, their functions during the outbreak of the pandemic are the most important gap in the literature in this field." The most important gap? Please justify this statement. Table 2 defines postmodern organizations. This term should be defined earlier in the paper.
Author Response

(The authors gave the same response as above.)

Reviewer 4 Report
Thank you for taking my suggestions and those of the other reviewers' into consideration.
Author Response

(The authors gave the same response as above.)

Reviewer 5 Report
The article has been greatly improved by including the recommendations of the reviewers.Congratulations!!
Author Response
Hi dear reviewer:
Thank you for taking the time to review the manuscript for our research team. We believe that your valuable comments can help improve our manuscript. Thank you so much for your positive message and comment.
sincerely yours

Round 3
Reviewer 3 Report
.